# SNAI2 Attenuated the Stem-like Phenotype by Reducing the Expansion of EPCAM^high^ Cells in Cervical Cancer Cells

**DOI:** 10.3390/ijms24021062

**Published:** 2023-01-05

**Authors:** Xian Liu, Ni Zhang, Qian Chen, Qian Feng, Yanru Zhang, Zhiqiang Wang, Xiong Yue, Hongbao Li, Nan Cui

**Affiliations:** 1Department of Reproductive Medicine, The First Affiliated Hospital of Xi’an Jiaotong University, Xi’an 710061, China; 2Section of Cancer Stem Cell Research, Key Laboratory of Environment and Genes Related to Diseases, Ministry of Education of the People’s Republic of China, Xi’an 710061, China; 3Department of Basic Medicine, Xi’an Medical University, Xin-Wang Street #1, Xi’an 710021, China; 4Department of Physiology and Pathophysiology, Xi’an Jiaotong University School of Basic Medical Sciences, Xi’an 710061, China; 5Gansu Provincial Maternity and Child-Care Hospital, Lanzhou 730050, China

**Keywords:** SNAI2, EPCAM, stem-like phenotype, β-catenin, cervical cancer

## Abstract

SNAI2 (Snai2) is a zinc-finger transcriptional repressor that belongs to the Snail family. The accumulated evidence suggests that SNAI2 exhibits biphasic effects on regulating a stem-like phenotype in various types of cells, both normal and malignant. In this study, by exogenously expressing SNAI2 in SiHa cells, SNAI2 exhibited the capacity to inhibit a stem-like phenotype in cervical cancer cells. The SNAI2-overexpressing cells inhibited cell growth, tumorsphere formation, tumor growth, enhanced sensitivity to cisplatin, reduced stem cell-related factors’ expression, and lowered tumor initiating frequency. In addition, the EPCAM^high^ cells sorted from SiHa cells exhibited an enhanced capacity to maintain a stem-like phenotype. Further study demonstrated that the trans-suppression of EPCAM expression by SNAI2 led to blockage of the nuclear translocation of β-catenin, as well as reduction in SOX2 and c-Myc expression in SiHa and HeLa cells, but induction in SNAI2 knockdown cells (CaSki), which would be responsible for the attenuation of the stem-like phenotype in cervical cancer cells mediated by SNAI2. All of these results demonstrated that SNAI2 could attenuate the stem-like phenotype in cervical cancer cells through the EPCAM/β-catenin axis.

## 1. Introduction

The Snail family is a highly conserved zinc-finger transcriptional family. SNAI2 (Snai2) is a member of this family, participating in numerous types of biological regulation processes, including epithelial–mesenchymal transition (EMT), cancer metastasis, and cellular reprogramming. More interestingly, SNAI2 has also been reported to play a specific role in the regulation of stem-cell-like properties in both normal and malignant cells, either by enhancing or attenuating the cell’s self-renewal capacity. By binding to YAP/TAZ, Snail/SNAI2 controlled self-renewal and differentiation in skeletal stem cells [1] and mesenchymal stem cells [2]. In head and neck squamous cell carcinoma (HNSCC), by facilitating sphere formation, inducing stem-cell-related factor expression, and enhancing chemoresistance to cisplatin, SNAI2 enhanced the self-renewal capacity of cells [3]. When SNAI2 and SOX9 are co-expressed in human lung carcinoma, SNAI2 can protect SOX9 avoid from ubiquitin-mediated proteasomal degradation, which is necessary for SOX9 to promote cancer stem cells and maintain its stem-like phenotype [4]. The co-operation of SOX9 and SNAI2 can also convert luminal cells into mammary stem cells (MaSCs) with long-term mammary gland reconstitution ability [5]. Surprisingly, mammary glands have been shown to develop in SNAI2-deficient mice, and functional mammary glands could be fully regenerated from the tissue fragments transplanted from these mice; thus, this could suggest that SNAI2 might have a negative capacity for stem cell activity [6]. In addition, SNAI2 also exhibited a dual role in regulating mammary epithelial cell (MEC) lineage identity, by promoting the stem cell transitions necessary for tumorigenesis. SNAI2 also repressed luminal epithelial differentiation [7]. Similarly, SNAI2 differentially regulates the potential for self-renewal in leukemia stem cells (LSCs), and normal hematopoietic stem cells (HSCs). During hematopoietic regeneration, SNAI2 deficiency enhanced HSC self-renewal occurs through the SCF/cKit signaling pathway by a novel negative feedback regulatory loop [8,9]. Moreover, SNAI2 promoted leukemogenesis, but SNAI2 deficiency impaired LSC self-renewal and delayed leukemia progression [10]. However, the potential role of SNAI2 in regulating the stem-like phenotype in cervical cancer cells is still unknown.

Epithelial cell adhesion molecule (EPCAM), which was firstly reported as a dominant antigen in human colon carcinoma as early as four decades ago, is a surface glycoprotein. The expression of EPCAM has been found in many types of stem cells, progenitor cells, epithelial tissues, and many types of cancers, and it is considered a multifunctional transmembrane protein, involved in the regulation of the cell stemness, growth, adhesion, and motility of cancer cells [11,12]. More importantly, in embryonic stem cells, EPCAM is required for stem cell survival or proliferation [13,14]. The EPCAM^high^ (or EPCAM^positive^) cell fraction within the liver has been described as a candidate progenitor cell population, and EPCAM has also been reported to be a cancer stem cell (CSC) biomarker for hepatocellular carcinoma [15,16,17,18]. CSCs are the small subset cells with the capacity to self-renew in cancer cells. The expression of EPCAM increases the expression of stemness markers (NANOG, SOX2, and OCT4) in cancer cells [19,20], and EPCAM has been described as a marker of cancer stem cells in colorectal, prostate, pancreatic, and breast cancers [21,22,23,24,25,26]. In cervical cancer, although it has been reported that inhibition of EPCAM expression significantly increases radiation-induced cell death in a cervical cell line (ME-180) [27], the potential correlation of EPCAM and the stem-like phenotype in cervical cancer cells remains unclear.

Our previous study demonstrated that SNAI2 could inhibit EPCAM expression by recognizing and directly binding the E-box motifs (CANNTG) in the proximal promoter region of EPCAM in cervical cancer cells, further inhibiting cell proliferation, and inducing EMT [28]. In this study, SNAI2 exhibited a capacity for inhibition of the stem-like phenotype in cervical cancer cell lines. Conversely, EPCAM^high^ cervical cancer cells exhibited an enhanced stem-like phenotype. Therefore, the central question is whether there is a potential interaction mechanism between SNAI2 and EPCAM in regulating the stem-like phenotype in cervical cancer cells.

## 2. Results

### 2.1. SNAI2 Inhibited the Stem-like Phenotype in Cervical Cancer Cells

In our previous study, SNAI2 exhibited a capacity to inhibit cell growth and tumor formation in cervical cancer cells [28,29]. In this study, the clonogenic formation assay showed that in both 2D (plate clone formation experiment, Figure 1A, *p* < 0.05) and 3D (soft agar assay, Figure 1B, *p* < 0.01) cell culture systems, the number and size of cell colonies were obviously decreased in SNAI2-overexpressing SiHa cells (SiHa-SNAI2), compared with control cells (SiHa-Vec). Moreover, SNAI2 was previously reported to play a mediator role in regulating the stem-like phenotype in several types of cancer [5,10]. To assess the effect of SNAI2 on the self-renewal ability of cervical cancer cells, SiHa-SNAI2 and SiHa-Vec cells were cultured in serum-free medium under optimal conditions for growing tumorspheres. As shown in Figure 1C, SiHa-SNAI2 cells generated fewer tumorspheres and fewer cell aggregates than the more conventional tumorspheres generated from SiHa-Vec cells. Moreover, tumorspheres were generated from SiHa-Vec cells with an efficiency of 12.13%, 8.725%, and 7.40% during three consecutive passages, whereas the tumorspheres were generated from SiHa-SNAI2 cells with an efficiency of 3.10%, 2.60% and 2.45% during three consecutive passages (Figure 1C).

Resistance to the chemotherapeutics of CSCs is one of the manifestations of cell self-renewal capacity [30]. Therefore, the effects of cisplatin in SNAI2-modified cells were tested in this study. After treatment with cisplatin for 24 h at different concentrations, cell viability was detected by using an MTT assay. As shown in Figure 1D, SiHa-SNAI2 cells were more sensitive to cisplatin treatment than SiHa-Vec cells when cisplatin concentrations were ≥6 µg/mL. Moreover, SiHa-SNAI2 cells were also more sensitive to cisplatin treatment than SiHa-Vec cells when cisplatin concentration was at 6 µg/mL and cultured for 24, 48, and 72 h (Figure 1E).

Stem-cell-related factors, such as SOX2, KLF4, NANOG, OCT4, ALDH1, and ALDH2, have been reported to be vital for maintaining cell self-renewal traits in many types of cancer. Therefore, RT–PCR was used to detect the expression of these stem-cell-related factors in SiHa-Vec and SiHa-SNAI2 cells. The mRNA levels of *SOX2*, *KLF4*, *NANOG*, *OCT4*, *ALDH1*, and *ALDH2* were decreased in SNAI2-overexpressing cells compared with control cells (Figure 1F, *p* < 0.05). Western blot analysis also confirmed the decrease in SOX2, KLF4, NANOG, OCT4, ALDH1, and ALDH2 expression in SiHa-SNAI2 cells (Figure 1G, *p* < 0.05). Moreover, decreased protein levels of SOX2, KLF4, NANOG, OCT4, ALDH1, and ALDH2 were also confirmed in SiHa-SNAI2-derived mouse xenografts (collected from our previous study) by Western blot [29] (Figure 1H, *p* < 0.05). All of these data indicated that overexpression of SNAI2 in cervical cancer cells attenuated the stem-like phenotype.

### 2.2. Exogenous Expression of SNAI2 Attenuated the Tumorigenic Capacity of Cervical Cancer Cells In Vivo

The powerful ability to initiate tumors was thought to be an important capacity of CSCs. In our previous study, SNAI2 exhibited an inhibitory effect on tumor formation in cervical cancer in a tumor-bearing model in nude mice [29]. To further determine the potential function of SNAI2 during tumor initiation, SNAI2-modified SiHa cells and their control cells were subcutaneously injected into NOD/SCID mice by limiting dilutions to 10, 100, 1000, and 10,000 cells, and then the tumor volume, tumor initiating frequency, and incidence were monitored. As shown in Figure 2A–D, the NOD/SCID mice that were subcutaneously injected with SiHa-Vec or SiHa-SNAI2 cells formed xenograft tumors at each cell concentration (10, 100, 1000, and 10,000 cells). However, the xenograft tumors formed by SiHa-SNAI2 cells were much smaller and grew much slower than the xenograft tumors formed by SiHa-Vec cells. In addition, the average weight of the xenograft tumors derived from SiHa-SNAI2 cells at each cell dose (10, 100, 1000 and 10,000 cells) was much lighter than the xenograft tumors derived from control cells (Figure 2E, *p* < 0.05).

Tumor latency was monitored after injection of SiHa-Vec and SiHa-SNAI2 cells into the NOD/SCID mice. As shown in Figure 2F, SiHa-SNAI2 cells exhibited a significantly longer tumor-free period than SiHa-Vec cells. In the SiHa-SNAI2 cell group, the shortest tumor-free period was 48 days, compared to 39 days in the SiHa-Vec cell group. The SiHa-SNAI2 cell group also exhibited a higher tumor-free rate (54.16% in SiHa-SNAI2 cells versus 4.17% in SiHa-Vec cells, *p* < 0.05) than the SiHa-Vec cell group.

The tumor incidence of SiHa-Vec and SiHa-SNAI2 cells is summarized in Figure 2G. The tumor-initiating frequency of SiHa-Vec was 1/286 at the lowest and 1/40.5 at the highest, compared to 1/7434 at the lowest (26.0-fold lower than SiHa-Vec cells, *p* < 0.05) and 1/1121.3 at the highest (27.7-fold lower than SiHa-Vec cells, *p* < 0.05) in SiHa-SNAI2. Therefore, the tumor formation assays revealed that the xenograft tumors derived from SiHa-SNAI2 cells grew much slower, and had a lower tumor initiating frequency, higher tumor-free rate, and longer tumor latency, compared with SiHa-Vec cells, suggesting that the tumorigenic capacity was attenuated by exogenous expression of SNAI2 in cervical cancer cells in vivo.

### 2.3. SiHa-EPCAM^high^ Cells Exhibit an Enhanced Stem-like Phenotype

In our previous study [28], SNAI2 exhibited the capacity to act as a transcriptional repressor of EPCAM in cervical cancer (Figure 3A). Western blot (Figure 3B) and immunocytochemical (Figure 3C) results also verified the decreased protein levels of EPCAM in SNAI2-overexpressing SiHa cells. Moreover, flow cytometry results also revealed a lower level of EPCAM-positive ratio in SiHa-SNAI2 cells compared with control cells (44.3% in SiHa-SNAI2 cells versus 93.4% in SiHa-Vec cells, Figure 3D).

As the advantageous effect of EPCAM on regulating self-renewal has previously been reported in various types of cancer cells, the potential role of EPCAM in regulating self-renewal in SiHa cells was confirmed in this study. First, EPCAM^high^ and EPCAM^low^ SiHa cells were sorted by FACS from SiHa cells (fluorescence-activated cell sorting, Figure 3E) and then cultured for generating tumorspheres. As shown in Figure 4F, EPCAM^high^ cells generated many more conventional tumorspheres than EPCAM^low^ cells. EPCAM^high^ cells generated tumorspheres with an efficiency of 15.53%, 14.25%, and 12.51% during three consecutive passages, whereas EPCAM^low^ cells generated tumorspheres with an efficiency of 3.50%, 4.36%, and 3.12% during three consecutive passages (Figure 3F).

To further determine the potential effect of EPCAM on regulating tumor initiation in cervical cancer cells, EPCAM^high^ and EPCAM^low^ cells were subcutaneously injected into NOD/SCID mice at limited dilutions with 10, 100, 1000, and 10,000 cells, respectively, and the tumor volume and incidence were monitored. As shown in Figure 3G, the NOD/SCID mice subcutaneously injected with EPCAM^high^ and EPCAM^low^ cells formed palpable tumors at each cell dose (10, 100, 1000, and 10,000 cells). However, the xenograft tumors formed by EPCAM^high^ cells grew much faster and to much larger sizes than the xenograft tumors formed by EPCAM^low^ cells. In addition, the average weight of the xenograft tumors derived from EPCAM^high^ cells at each cell dose (10, 100, 1000, and 10,000 cells) were much heavier than the xenograft tumors which derived from EPCAM^low^ cells (Figure 3H, *p* < 0.05).

Moreover, as shown in Figure 3I, EPCAM^high^ cells exhibited a significantly shorter tumor-free period than EPCAM^low^ cells. A shorter tumor-free period was observed in EPCAM^high^ cells compared to EPCAM^low^ cells. In EPCAM^high^ cells, the shortest tumor-free period was 29 days, compared to 41 days in EPCAM^low^ cells. EPCAM^high^ cells also exhibited a lower tumor-free rate (8.33%) compared to EPCAM^high^ cells (41.67%, *p* < 0.05).

The tumor initiation frequency of EPCAM^high^ and EPCAM^low^ cells is summarized in Figure 3J. The tumor initiation frequency of EPCAM^high cells^ was 1/25.9 at lower levels (44.9-fold lower than EPCAM^low^ cells, *p* < 0.05) and 1/3.78 at higher levels (53.9-fold lower than EPCAM^low^ cells, *p* < 0.05), and the tumor-initiating frequency of EPCAM^low^ cells was 1/1162.4 at lower levels and 1/200.12 at higher levels. Therefore, the tumor formation assays from NOD/SCID mice revealed that EPCAM^high^ cells grew much faster, had a higher tumor initiating frequency, lower tumor-free rate, and shorter tumor latency compared to EPCAM^low^ cells, suggesting that EPCAM^high^ cells have a potential ability to promote tumor initiation and growth in vivo. All of these results suggested that EPCAM^high^ cervical SiHa cells exhibit enhanced self-renewal capacity compared with EPCAM^low^ cells, and that EPCAM expression is advantageous for maintaining self-renewal capacity in SiHa cells.

### 2.4. Absence of EPCAM Inhibited the Nuclear Translocation of β-Catenin in SNAI2-Overexpressing Cervical Cancer Cells

In order to explore whether the alteration of EPCAM under SNAI2-overexpressing cells was involved in SNAI2 mediating the stemness of cervical cancer cells, EPCAM^high^ and EPCAM^low^ cells were sorted by FACS from SiHa-SNAI2 cells (Figure 4A), and then cultured to generate tumorspheres. As shown in Figure 4B, EPCAM^high^ cells from SiHa-SNAI2 cells generated many more tumorspheres than EPCAM^low^ cells from SiHa-SNAI2 cells. SiHa-SNAI2-EPCAM^high^ cells generated tumorspheres with an efficiency of 6.13%, 7.75%, and 6.91% during three consecutive passages, whereas SiHa-SNAI2-EPCAM^low^ cells generated tumorspheres with an efficiency of 3.25%, 2.66%, and 2.59% during three consecutive passages (Figure 4B). Moreover, an EPCAM recombinant plasmid was used to rescue the expression of EPCAM in SiHa-SNAI2 cells. As shown in Figure 4C, SiHa-SNAI2-EPCAM cells generated many more tumorspheres than SiHa-SNAI2-Vec cells. SiHa-SNAI2-EPCAM generated tumorspheres with an efficiency of 7.53%, 6.25%, and 4.15% during three consecutive passages, whereas the SiHa-SNAI2-Vec cells generated tumorspheres was with an efficiency of 2.65%, 1.86%, and 1.79% during three consecutive passages (Figure 4C). These results suggest that EpCMA is involved in the SNAI2-mediated stem-like phenotype in cervical cancer cells.

The Wnt/β-catenin signaling pathway plays a critical role in promoting cancer progression and maintaining cell stemness, not only in embryonic stem cells but also in many types of cancer stem cells. The interaction between EPCAM and the Wnt/β-catenin signaling pathway for maintaining cell stemness was reported in a previous study [31]. Our previous study demonstrated that SNAI2 attenuated the activity of the Wnt/β-catenin signaling pathway in cervical cancer cells [29]. Therefore, the potential interactions between SNAI2, EPCAM, and the Wnt/β-catenin signaling pathway were investigated in this study. First, the protein levels of β-catenin and the Wnt/β-catenin signaling pathway target genes c-Myc and SOX2 were detected in SNAI2-modified cervical cancer cells. As shown in Figure 4D,E, the protein levels of β-catenin, c-Myc, and SOX2 were decreased in SNAI2-overexpressing SiHa and HeLa cells compared with the control cells (SiHa-Vec and HeLa-Vec cells, *p* < 0.05). Conversely, the protein levels of β-catenin, c-Myc, and SOX2 were increased in SNAI2-knockdown CaSki-shSNAI2 cells compared with control cells (CaSki-shCtr cells, *p* < 0.05).

EPCAM comprises an extracellular domain (EpEX) and an intracellular domain (EpICD) [11]. EpICD becomes part of a large nuclear complex containing the transcriptional regulator β-catenin, which is an important component of the Wnt/β-catenin signaling pathway, and enhances the nuclear translocation of β-catenin [32]. Therefore, the protein level of β-catenin in the nucleus was detected in this study. As shown in Figure 4D, the protein level of β-catenin in the nucleus was decreased in SNAI2-overexpressing SiHa and HeLa cells compared with the control cells (SiHa-Vec and HeLa-Vec cells, *p* < 0.05). Conversely, the protein level of β-catenin was increased in SNAI2-knockdown CaSki-shSNAI2 cells compared with control cells (CaSki-shCtr cells, *p* < 0.05).

To further confirm that the reduction in c-Myc and SOX2 in SNAI2-overexpressing cells could be attributed to the attenuated activity of the Wnt/β-catenin signaling pathway, the reduction in β-catenin was recovered by transiently transfecting a recombinant β-catenin plasmid. As shown in Figure 4E, the expressions of β-catenin, c-Myc, and SOX2 were recovered by transfection with pIRES2-AcGFP-β-catenin into SiHa-SNAI2 and HeLa-SNAI2 cells. Conversely, XAV939, which is an inhibitor of the Wnt/β-catenin signaling pathway, was used to block the stimulated β-catenin expression in CaSki-shSNAI2 cells. As shown in Figure 4F, after treatment with XAV939, the expressions of β-catenin, c-Myc, and SOX2 were reduced in CaSki-shSNAI2 cells. These results demonstrated that the alteration of c-Myc and SOX2 in SNAI2-modified cells could be attributed to changes in the Wnt/β-catenin signaling pathway.

Furthermore, when the protein level of EPCAM was recovered in SNAI2-overexpressed cells, β-catenin, c-Myc, SOX2, and nuclear β-catenin expression were also recovered in SNAI2-overexpressed SiHa and HeLa cells (Figure 4G,H, *p* < 0.05). Similarly, the same phenomenon was observed in SiHa-Vec and HeLa-Vec cells by transiently transfecting the EPCAM recombinant plasmid (Figure 4G,H, *p* < 0.05). All of these results suggested that SNAI2 blocked the nuclear translocation of β-catenin by inhibiting EPCAM expression, further suppressing the expression of c-Myc and SOX2, and ultimately attenuating the stem-like phenotype in cervical cancer cells.

### 2.5. SNAI2 Exhibited a Negative Correlation with EPCAM in Cervical Cancer Samples In Vivo

Additionally, the protein levels of SNAI2, EPCAM, β-catenin, c-Myc, and SOX2 were detected in xenografted tumor samples formed by SiHa-Vec and SiHa-SNAI2 cells. As shown in Figure 5A,C, SNAI2 exhibited a much higher protein level in SiHa-SNAI2 cell-derived xenografted tumor tissues than in SiHa-Vec cell-derived xenografted tumor tissues (*p* < 0.05). Additionally, lower EPCAM, β-catenin, β-catenin (in the nucleus), c-Myc, and SOX2 protein levels were seen in SiHa-SNAI2-cell-derived xenografted tumor tissues compared to SiHa-Vec-cell-derived xenografted tumor tissues (*p* < 0.05).

Moreover, the protein levels of SNAI2, EPCAM, β-catenin, c-Myc, and SOX2 were also detected in EPCAM^high^-and EPCAM^low^-cell-derived xenografted tumor tissues. As shown in Figure 5B,D, higher expressions of EPCAM, β-catenin, β-catenin (in the nucleus), c-Myc, and SOX2 were observed in the EPCAM^high^ cells derived xenografted tumor tissues than in the EPCAM^low^ cell-derived xenografted tumor tissues (*p* < 0.05). In particular, an increased protein level of SNAI2 was observed in EPCAM^high^-cell-derived xenografted tumor tissues than in the EPCAM^low^-cell-derived xenografted tumor tissues (*p* < 0.05). This could be attributed to the activation of the Wnt/β-catenin signaling pathway under higher EPCAM expression. As reported in a previous study, activation of the Wnt/β-catenin signaling pathway enhanced SNAI2 expression in various cancers [33].

Additionally, the SNAI2 and EPCAM protein levels in human CESC specimens were detected using the TMA (tissue microarray) approach. The selected CESC TMA included 13 adenocarcinoma of the uterine cervix specimens (AUC), and 13 squamous carcinomas of the cervix specimens (SCC). The relationship between EPCAM and SNAI2 in these CESC specimens was examined via Pearson correlation analysis by performing immunohistochemistry (IHC). Representative images of high and low staining of SNAI2 and EPCAM are shown in Figure 5E. A negative correlation between the expression of EPCAM and SNAI2 was observed in all 26 CESC specimens (Figure 5E, r = −0.7393, *p* = 0.0008) and SCC (n = 13, Figure 5E, r = −0.7564, *p* = 0.0028) specimens, respectively, which was consistent with our previous study and the data from the GEPIA online database [29]. Furthermore, the IRSs (immunoreactive scores) of SNAI2 and EPCAM in AUC and SCC specimens are shown in Figure 5G. The mean IRS value for SNAI2 in SCC specimens (n = 13, 7.462 ± 4.115) was much higher than that in AUC specimens (n = 13, 1.385 ± 1.445, *p* < 0.05). The mean IRS for EPCAM in SCC specimens (n = 13, 6.000 ± 3.266) was much lower than that in AUC specimens (n = 13, 9.769 ± 3.270, *p* < 0.05). Together, these results suggest that SNAI2 is negatively correlated with EPCAM in human CESC patient specimens.

## 3. Discussion

SNAI2 is well known as an EMT regulator in various carcinomas. The accumulated evidence suggests that SNAI2 exhibits biphasic effects on regulating, attenuating, or enhancing the stem-like phenotype. In our previous study, SNAI2 exhibited the capacity to inhibit cell proliferation, and promote cell motility and distant metastasis by trans-suppressing EPCAM expression in cervical cancer [28]. In addition, the results of TMA (tissue microarray) also confirmed the negative correlation of SNAI2 and EPCAM protein level in cervical cancer tissues. In this study, SNAI2 exhibited the capacity to inhibit a stem-like phenotype in cervical cancer. As shown in Figure 1, by exogenously expressing SNAI2 in SiHa cells, SiHa-SNAI2 cells exhibited inhibited cell growth, decreased tumorsphere formation, enhanced sensitivity to cisplatin, reduced stem-cell-related factor expression, a slower tumor growth rate, and a lower tumor initiating frequency, indicating that SNAI2 plays an inhibitory role in regulating a stem-like phenotype in cervical cancer cells.

EPCAM is considered a multifunctional transmembrane protein involved in the regulation of cell stemness and is required for stem cell survival or proliferation, and the inhibitory role of SNAI2 on trans-suppressing EPCAM expression in cervical cancer was reported in our previous study [28]. Thus, in cervical cancer, there is a potential connection between SNAI2 trans-suppression of EPCAM expression and SNAI2-mediated inhibition of the stem-like phenotype. Although the advantageous effect of EPCAM on regulating cell stemness has been reported in various cancers, it has been rarely studied in cervical cancer. As shown in Figure 3, the EPCAM^high^ cells sorted from SiHa cells generated many more conventional tumorspheres, grew much faster, and had a higher tumor initiating frequency, shorter tumor latency, and lower tumor-free rate. These results indicated that EPCAM expression facilitated the maintenance of cell renewal in SiHa cells. Moreover, both the EPCAM^high^ cells sorted from SiHa-SNAI2 cells, or transiently transfected with an EPCAM recombinant plasmid to rescue EPCAM expression in SiHa-SNAI2 cells, exhibited enhanced cell-renewal ability, suggesting that alteration of EpCMA is involved in the SNAI2-mediated stem-like phenotype in cervical cancer cells.

EPCAM comprises an extracellular domain (EpEX) and an intracellular domain (EpICD) [11]. EpICD becomes part of a nuclear complex containing β-catenin, and thus is involved in the nuclear translocation of β-catenin [32]. In this study, SNAI2 overexpression inhibited the expression of EPCAM, β-catenin, and nuclear β-catenin in SiHa and HeLa cells (Figure 4). Conversely, when EPCAM expression was rescued in SNAI2-overexpressing cells, the protein levels of EPCAM, β-catenin, and nuclear β-catenin were recovered in SNAI2-overexpressing cells. In general, blockade of the nuclear translocation of β-catenin attenuated the activity of the Wnt/β-catenin signaling pathway [34]. These findings are in accordance with our previous study, which demonstrated that SNAI2 attenuated the activity of the Wnt/β-catenin signaling pathway in cervical cancer cells [29]. All of these results suggested that SNAI2 could inhibit the nuclear translocation of β-catenin by trans-suppressing EPCAM expression, further attenuating the activity of the Wnt/β-catenin signaling pathway.

The Wnt/β-catenin signaling pathway plays a critical role in promoting cancer progression and maintaining cell stemness, not only in embryonic stem cells but also in many types of cancer stem cells. SOX2 and c-Myc are considered target genes of the Wnt/β-catenin signaling pathway, and both are also necessary for maintaining cell stemness [35,36,37,38]. In this study, a reduction in SOX2 and c-Myc expression was observed in SNAI2-overexpressing SiHa and HeLa cells. Conversely, when the expression of β-catenin or EPCAM was rescued in SNAI2-overexpred cells, the protein levels of SOX2 and c-Myc were recovered in SNAI2-overexpressing cells. Additionally, the negative correlation between SNAI2 and EPCAM, β-catenin, SOX2, and c-Myc expression in xenograft tissues from NOD/SCID mice (derived from SiHa-Vec and SiHa-SNAI2 cells) was confirmed in this study.

In conclusion, this study reveals a negative relationship between SNAI2 and EPCAM in cervical cancer. Moreover, this study suggested that blocking the nuclear translocation of β-catenin and further reducing SOX2 and c-Myc expression, which contribute to the trans-suppression of EPCAM expression by SNAI2, are responsible for attenuation of the stem-like phenotype in cervical cancer cells mediated by SNAI2 (Figure 6).

## 4. Materials and Methods

### 4.1. Cell Lines and Cell Culture

The cervical cancer cell lines that were used in this study were procured from ATCC (Manassas, VA, American Type Culture Collection), including HeLa, CaSki, and SiHa.

The cell lines were authenticated by STR profiling and RT-PCR was used to test mycoplasma contamination. The cell lines that stably overexpressed (SiHa-SNAI2, HeLa-SNAI2) or knockdown of SNAI2 (CaSki-shSNAI2) and control cell lines (SiHa-Vec, HeLa-Vec, CaSki-shCtr) were generated and saved in our laboratory [29]. DMEM (Sigma-Aldrich, St. Louis, MO, USA) with 10% FBS (HyClone, Thermo Scientific, Waltham, MA, USA) and G418 (Calbiochem, La Jolla, CA, USA) was used to culture SiHa and HeLa cells. RPMI-1640 medium (Sigma-Aldrich, St. Louis, MO, USA) with 10% FBS and G418 was used to culture CaSki cells. All of the cells were incubated at 37 °C in a humidified 5% CO_2_ atmosphere.

### 4.2. Colony Formation Assay

The colony formed assay was used to evaluate the capacity of long-term cell proliferation and survival in 2D culture. In brief, SiHa-GFP and SiHa-SNAI2 cells were seeded as the cell density of 500 cells/well into 6-well plates, respectively. Then, the cells were in culture for 2 weeks. At the end of this experiment, the cells were washed by using PBS, fixed with paraformaldehyde (4%, for 15 min), then stained using crystal violet (0.5%, w/v, Sigma-Aldrich, St. Louis, MO, USA). The ImageJ (version 1.48 v) software was used to count the number of colonies [39].

### 4.3. Soft Agar Assay

The soft agar assay was used to evaluate the capacity of long-term cell proliferation and survival in 3D culture. The culture solution that contained 0.5% low-melting agarose was plated in the lower-layer of the six-well plates, then cultured at 37 °C for 4–5 h. The upper-layer with low-melting agarose to a final concentration of 0.3% was mixed with 2000 cells/well and added to 500μL of the culture medium, then cultured for 2–4 weeks at 37 °C. The ImageJ (version 1.48v) software was used to count the number of colonies.

### 4.4. Real-Time PCR Analysis

The experimental procedure for extracting the total RNA was followed as described in our previously study [29]. Additionally, the experimental procedures for RT-PCR were implemented as per the description of the reagent supplies’ manual and a previous study [40]. The ∆∆Ct method was used to analyze the results, and the house-keeping gene was GAPDH in this study. The primer sequences which performed real-time PCR in this study were as follows: *SOX2* (F: 5′-TACAGCATGTCCTACTCGCAG-3′ R: 5′-GAGGAAGAGGTAACCACAGGG-3′), *KLF4* (F: 5′-CCCACATGAAGCGACTTCCC-3′ R: 5′-CAGGTCCAGGAGATCGTTGAA-3′), *NANOG* (F: 5′- TTTGTGGGCCTGAAGAAAACT-3′ R: 5′-AGGGCTGTCCTGAATAAGCAG -3′), *Oct4* (F: 5′- CTTGAATCCCGAATGGAAAGGG-3′ R: 5′-: TGTATATCCCAGGGTGATCCTC -3′), *ALDH1* (F: 5′-: GTGTATATCCCAGGGTGATCCTC -3′ R: 5′-CCTCCTCAGTTGCAGGATTAAAG -3′), *ALDH2* (F: 5′- ATGGCAAGCCCTATGTCATCT-3′ R: 5′-CCGTGGTACTTATCAGCCCA -3′), *SNAI2* (F: 5′- CGAACTGGACACACATACAGTG-3′ R: 5′-CTGAGGATCTCTGGTTGTGGT -3′) and *GAPDH* (F: 5′-CACCGTCA AGGCTGAGAAC-3′ and 5′-TGGTGAAGACGCCAGTGGA-3′).

### 4.5. Tumorsphere Culture

The cells were cultured in stem cell media; this medium was constituted with the DMEM/F12 basal media, and added with hEGF (EGF, 20 ng/mL), N2 (Invitrogen), bFGF (20 ng/mL, PeproTech Inc., Rocky Hill, NJ, USA), and B27 (Invitrogen) supplements. The cells were plated as 1 cell (per well) in 96-well plates and cultured in this stem cell medium for the tumorsphere formation assay. After 2–3 weeks, the number of tumorspheres were counted. The tumorspheres were harvested and disaggregated with 0.25% trypsin/EDTA, then plated as 1 cell (per well) in 96-well plates, and cultured in this stem cell medium for serial tumorsphere formation assays.

### 4.6. Cisplatin Resistance and MTT Assay

For cisplatin sensitivity assays, 10^4^ cells per well were plated in 96-well plates and the various concentrations of cisplatin were added (the concentrations of cisplatin were 0, 3, 6, 12, 24, 48, or 96 µg/mL) the next day. Then, the cells were exposed for 24 h, and the viability of such cells was measured by using MTT assays. Additionally, in a separate experiment, the cells were cultured at 6 µg/mL for 24, 48, and 72 h as a constant concentration of cisplatin, and then the MTT assays was used to detect the cell viability. The experimental procedure for MTT was followed as described in our previous study [28].

### 4.7. Western Blotting

The experimental procedure for Western blotting analysis that was used in this study was followed as described in our previous study [29]. The horseradish peroxidase-conjugated IgG that was anti-mouse and anti-rabbit (Thermo Fisher Scientific, New York, NY, USA) was used in this study. GAPDH was used as the control and for quantification. Antibodies purchased from Santa Cruz, USA: anti-EPCAM (1:1000, sc-25308), anti-ALDH1 (1:300, sc-374149), anti-ALDH2 (1:300, sc-100496), anti-KLF4 (1:1000, sc-166100), anti-GAPDH (1:1000, sc-47724). Antibodies purchased from Cell Signaling Technology, USA: anti-SNAI2 (#9585, 1:1000), anti-SOX2 (#3579, 1:1000), anti-OCT4 (#2840, 1:1000), anti-NANOG (#4903, 1:1000), anti-β-catenin (#8480, 1:1000), anti-c-Myc (#18583, 1:1500 dilution). Anti-PCNA (#60097-1-Ig, 1:5000) were purchased from Proteintech. The signal intensity was quantified using the protein imprinting imaging system (Tanon 5200, Better Tanon, Shanghai, China).

### 4.8. Tumor Xenograft Assay

The SiHa-Vec, SiHa-SNAI2, EPCAM^high^, and EPCAM^low^ cells were resuspended in 200 μL culture solution, which was mixed with DMEM/F12 basal media and Matrigel (#356234, BD, Biosciences) at 1:1, respectively. Then, it was injected subcutaneously into the female NOD/SCID mice (6 to 8-weeks old): left (SiHa-Vec or EPCAM^low^ cells) and right (SiHa-SNAI2 or EPCAM^high^ cells). All of the mice were fed in the SFP room, and the mice were observed every 3 days, and the length (a) and the width (b) of the tumors were recorded. The volume of the tumors was calculated as V = abb/2.

At the end of this experiment, all of the tumor samples were collected, weighed, and immunostained. The limiting-dilution analysis was used to evaluate the upper and lower limits of tumor formation, which represented the frequency of tumorigenic cells [41].

### 4.9. Plasmid and Transfection

The pIRES2-EPCAM [28] and pIRES2-AcGFP-β-catenin [42] plasmids were kept in our laboratory. In this study, the Lipofectamine 2000 reagent (Invitrogen, Carlsbad, CA, USA) was used to transiently transfect the plasmids in the cells that were used in our study. Additionally, the protocol was performed as per the description of the reagent supplies manual. Then, the proteins of these cells were collected and used for Western blotting.

### 4.10. EPCAM^high^ Cells and EPCAM^low^ Cells Isolation and Flow Cytometry Analysis

The antibody of EPCAM (1:100, sc-25308, Santa Cruz, CA, USA) in accordance with manufacturer’s instructions were used for flow cytometry and FACS to measure the expression of EPCAM in human cervical cells and xenografts, and sort EPCAM^high^ cells and EPCAM^low^ cells. Flow cytometry was performed with a FACS Calibu, and FACS was performed with a BD FACSAria II cell sorter (Becton Dickinson). The data were analyzed by FlowJo 7.6 software (Tree Star Inc., Ashland, OR, USA).

### 4.11. Immunocytochemistry, Immunohistochemistry (IHC) and Tissue Microarray

The experimental procedures for immunocytochemistry and immunohistochemistry that were used in this study were followed as described in our previous study [28]. The cervical cancer tissue microarray (Cat No. F261301, Bioaitech, Xi’an, China) was used to analyze the correlation between the expression of EPCAM (1:100 dilution) and SNAI2 (1:50 dilution). Then, the IHC score of EPCAM and SNAI2 was analyzed by the Pearson correlation analysis (all of the data have been tested for normality of distribution using GraphPad Prism 9.0).

### 4.12. Statistical Analysis

SPSS (version 19.0, Inc., Chicago, IL, USA) and GraPhpad Prism 9.0 (GraphPad Software, LLC, USA) software was used for statistical analysis of the data in this study. Continuous data were compared using Student’s *t*-tests, and data are expressed as mean ± SD. Pearson correlation analysis was used to analyze the correlation among protein expression. Survival free interval curves were calculated according to the Kaplan–Meier method, and analysis was performed using the logrank test.

## Figures and Tables

**Figure 1 ijms-24-01062-f001:**
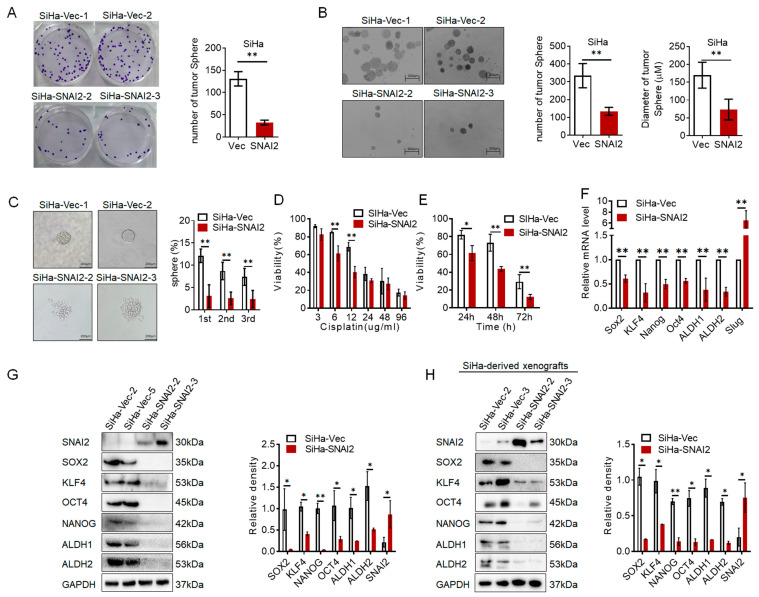
SNAI2 attenuated a stem-like phenotype in cervical cancer. (**A**). The colony formation assay was used to detect the long-term cell survival and growth of SNAI2-overexpressed SiHa-SNAI2 cells and control cells (SiHa-Vec). (**B**). The soft agar assay was used to detect the long-term cell survival and growth in 3D culture of SiHa-SNAI2 and SiHa-Vec cells. (**C**). The number of tumorspheres cells formed by SiHa-SNAI2 and SiHa-Vec cells was measured from three consecutive passages; the representative photos are shown. (**D**). MTT assay was used to detect the cell viability of SiHa-SNAI2 and SiHa-Vec cells by treating with different concentrations of cisplatin (3, 6, 12, 24, 48, 96 µg/mL) for 24 h. (**E**). MTT assay was used to detect the cell viability of SiHa-SNAI2 and SiHa-Vec cells by treating with a constant dose of cisplatin (6 µg/mL) for 0, 24, 48, or 72 h. (**F**). The mRNA level of SNAI2, SOX2, KLF4, NANOG, Oct3/4, ALDH1, and ALDH2 in SNAI2-modified cells was detected by real-time quantitative PCR. The protein level of SNAI2, SOX2, KLF4, NANOG, Oct3/4, ALDH1, and ALDH2 in SNAI2-modified cells (**G**) or SNAI2-modified cells derived mouse xenografts (**H**) was detected by western blotting. * *p* < 0.05, ** *p* < 0.01 vs. control.

**Figure 2 ijms-24-01062-f002:**
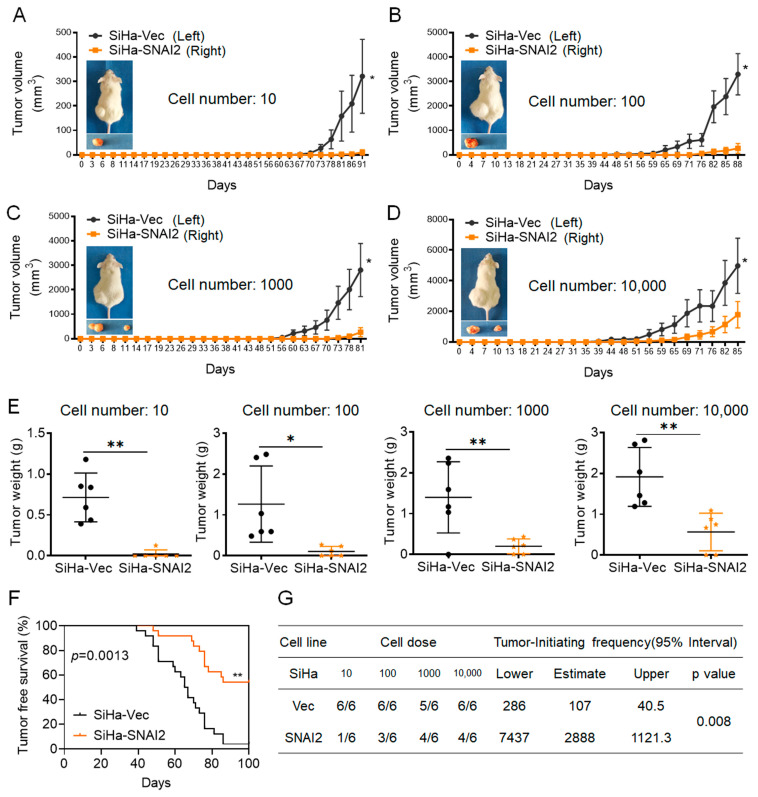
Tumorigenicity of SNAI2-modified SiHa cells in in NOD/SCID mice. The volume of xenograft tumors that formed from different numbers of SiHa-SNAI2 and SiHa-Vec cells: (**A**). 10, (**B**). 100, (**C**). 1000, (**D**). 10,000. (**E**). The weight of xenograft tumors that formed from different numbers of SiHa-SNAI2 and SiHa-Vec cells. (**F**). The tumor-free survival of the mice shown as Kaplan–Meier plots. (**G**). The tumor initiating frequency of SiHa-SNAI2 and SiHa-Vec cells. * *p* < 0.05, ** *p* < 0.01 vs. control.

**Figure 3 ijms-24-01062-f003:**
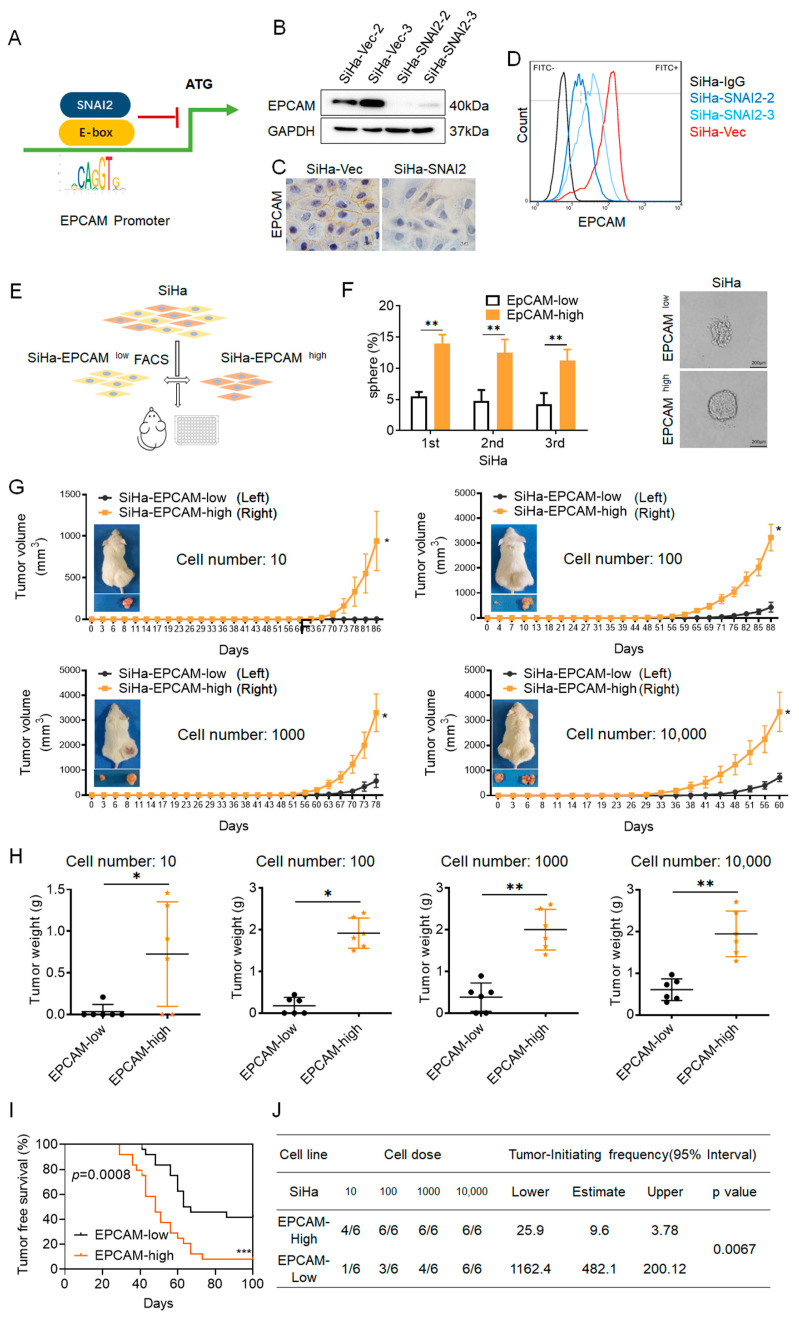
EPCAM^high^ cells exhibited enhanced stem-like phenotype. (**A**). The transcription inhibition pattern of SNAI2 for EPCAM. The expression of EPCAM in SiHa-SNAI2 and SiHa-Vec cells was detected by Western blotting (**B**), immunocytochemical (**C**) and flow cytometry (**D**). (**E**). The EPCAM^high^ and EPCAM^low^ SiHa cells were sorted by FACS from SiHa cells. (**F**). The number of tumorsphere cells formed by EPCAM^high^ and EPCAM^low^ SiHa cells was counted from three consecutive passages; the representative photos and the quantitative analysis are shown. (**G**). The volume of xenograft tumors that formed from different numbers of EPCAM^high^ and EPCAM^low^ SiHa cells. (**H**). The weight of xenograft tumors that formed from different numbers of EPCAM^high^ and EPCAM^low^ SiHa cells. (**I**). The tumor-free survival of mice that were injected with EPCAM^high^ and EPCAM^low^ SiHa cells shown as Kaplan–Meier plots. (**J**). The tumor initiating frequency of EPCAM^high^ and EPCAM^low^ SiHa cells. * *p* < 0.05, ** *p* < 0.01, *** *p* < 0.001 vs. control.

**Figure 4 ijms-24-01062-f004:**
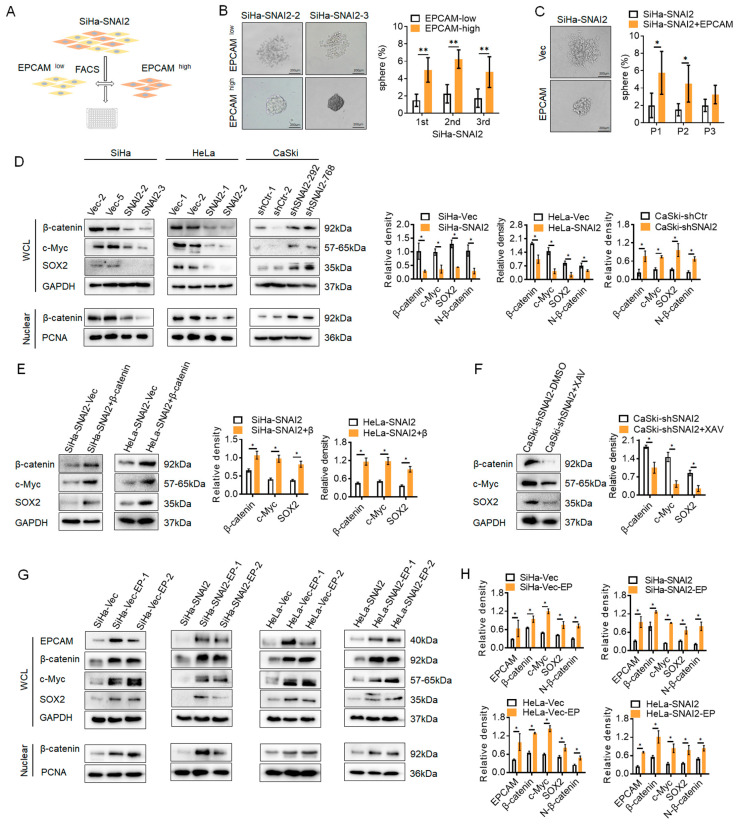
SNAI2 suppressed the nuclear translocation of β-catenin. (**A**). The EPCAM^high^ and EPCAM^low^ cells were sorted by FACS from SiHa-SNAI2 cells. (**B**). The number of tumorspheres formed by SiHa-SNAI2-EPCAM^high^ and SiHa-SNAI2-EPCAM^low^ cells was counted during three consecutive passages, and the representative photos are shown. (**C**). An EPCAM recombinant plasmid was used to recover the EPCAM expression in SiHa-SNAI2 cells. The number of tumorspheres formed by SiHa-SNAI2 and SiHa-SNAI2-EPCAM cells was counted during three consecutive passages, and the representative photos are shown. (**D**). The protein levels of β-catenin, c-Myc, SOX2, and nuclear β-catenin in SNAI2-modified cells were detected by Western blotting. (**E**). Western blotting was used to detect the protein levels of β-catenin, c-myc, and SOX2 after the β-catenin expression was rescued via transient transfecting of a β-catenin recombinant plasmid in SiHa-SNAI2 cells. (**F**). β-catenin expression was blocked by using XAV939 (60 µM/mL) in CaSki-shSNAI2 cells, and the protein levels of β-catenin, c-myc, and SOX2 were detected by Western blotting. (**G**). EPCAM expression was rescued via transient transfection of an EPCAM recombinant plasmid in SNAI2 overexpressed cells, and the protein levels of β-catenin, c-Myc, SOX2, and nuclear β-catenin were detected by Western blotting, and the quantitative analysis is shown in (**H**). * *p* < 0.05, ** *p* < 0.01, vs. control.

**Figure 5 ijms-24-01062-f005:**
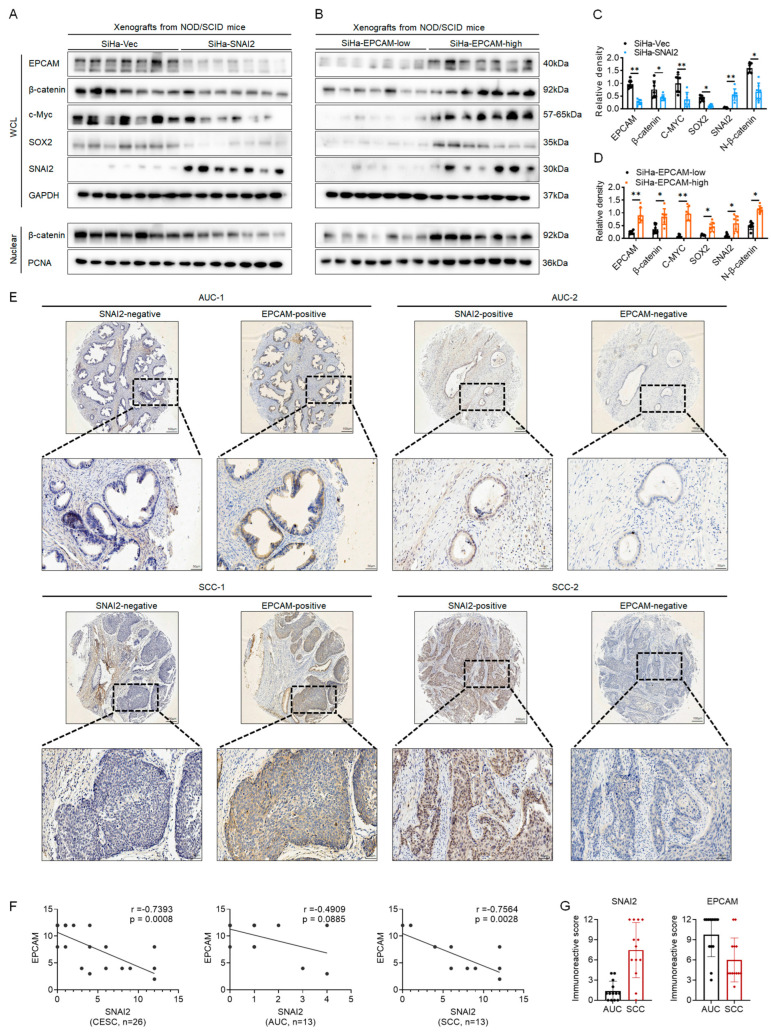
The negative correlation of the protein level of SNAI2 and EPCAM in tissue samples. The protein levels of EPCAM, β-catenin, c-Myc, SOX2, SNAI2, and nuclear β-catenin in SiHa-SNAI2- and SiHa-Vec-cells derived mouse xenografts (**A**,**C**) or in SiHa-EPCAM^high^- and SiHa- EPCAM^low^-cells-derived mouse xenografts (**B**,**D**) were detected by Western blotting. (**E**). The protein levels of SNAI2 and EPCAM in human tissue were detected via using the TMA (tissue microarray) approach, and the representative photos are shown. Scale bars, 400× and 1000×. (**F**). The Pearson correlation analysis was performed to analyze the correlation between SNAI2 and EPCAM in CESC (n = 26), adenocarcinoma of the uterine cervix specimens (AUC, n = 13), and squamous carcinoma of the cervix specimens (SCC, n = 13) samples. (**G**). The immunoreactive score (IRS) of SNAI2 and EPCAM in AUC and SCC specimens. * *p* < 0.05, ** *p* < 0.01, vs. control.

**Figure 6 ijms-24-01062-f006:**
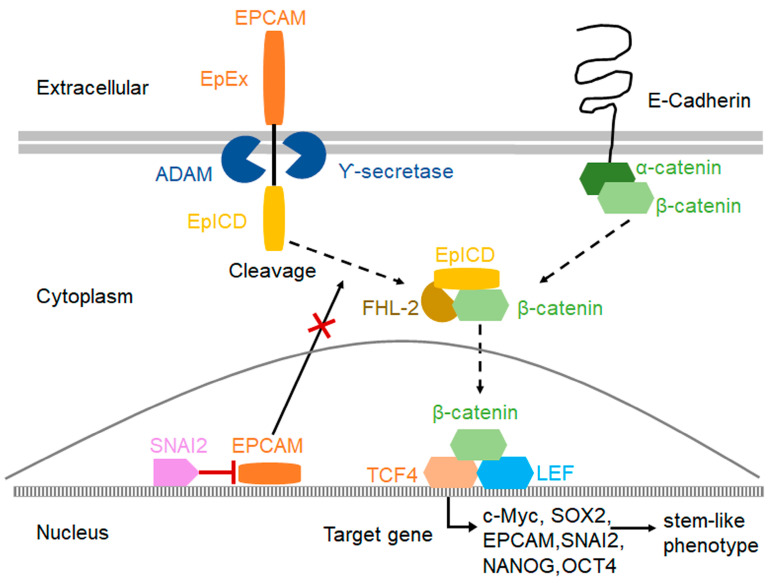
The proposed model of the mechanisms by which SNAI2 inhibits EPCAM expression and further attenuates the stem-like phenotype in cervical cancer cells.

## Data Availability

Not applicable.

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
