# Peer review of "SNAI2 Attenuated the Stem-like Phenotype by Reducing the Expansion of EPCAMhigh Cells in Cervical Cancer Cells"

_ijms, 2023, doi:10.3390/ijms24021062_

Round 1
Reviewer 1 Report
In this study, LIU et al. focus on the role of Slug factor transcription in his capacity to inhibit a stem-like phenotype in cervical cancer. The manuscript submitted here describes series of results clearly presented. They use successfully both in vitro studies and in vivo xenograft approaches. The authors observed that Slug overexpression exhibited an inhibition of cell growth, of tumorspheres formation and enhanced sensitivity to Cisplatin. Additionally, Slug overexpression induced reduction of stem cell-related factors expression as Sox2, Oct4, Nanog and Epcam. Using in vivo tumorigenesis approach, authors identified that Slug surexpression diminished tumor growth and elegantly studied crrelation between b cathenin signaling pathway with Slug and Epcam expression.
The quality of writing was good. Flow Cytometry and Immunoblot quality was fine and the technical merit was very good. Results are elegantly shown. Technical terms were rigorously and well detailed.
Overall, the study is within the scope of IJMS
Author Response
Response: Thank you for your suggestion. We feel very honored for your approbation of this study. Thank you very much.
Reviewer 2 Report
In this manuscript, the author discovered that Slug inhibited the stem-like phenotype of SiHa cells through the EpCAM/b-catenin axis. Overall, this is a comprehensive study that identified a new suppressing factor for cervical cancer, along with the potential signaling pathway. I still have several questions:
1. The authors proposed that the Slug, as a zinc finger transcriptional factor, would bind to the EpCAM promoter. Yet they didn’t provide any experimental evidence. The author should at least use luciferase assay to test whether Slug binds to the EpCAM promoter and suppresses the expression.
2. In the proposed model, Slug suppressed the translocation of b-catenin. However, based on the WB results, the amount of b-catenin decreased significantly in both WCL and nuclear portions. Slug likely suppressed the expression of b-catenin rather than its translocation into the nuclear.
3. Most of the experiments in this study are based on the overexpression of Slug in SiHa cells. I wonder about the expression level of Slug in cancer tissues compared to normal tissue. Would the cancer cell express less Slug? I also found a previous study reporting that Slug promotes breast cancer progression (PMID: 25652255). Does that mean that Slug's regulatory role differs in different cancer cells?
4. I appreciate the model diagram in Figure 6. Yet some part of the model is hypothetical and without any experimental evidence. The authors should modify this diagram by adding references or labeling unverified parts in dash lines/ semi-transparent to avoid misinformation.
Author Response
Comments and Suggestions for Authors
In this manuscript, the author discovered that Slug inhibited the stem-like phenotype of SiHa cells through the EpCAM/b-catenin axis. Overall, this is a comprehensive study that identified a new suppressing factor for cervical cancer, along with the potential signaling pathway. I still have several questions:
- The authors proposed that the Slug, as a zinc finger transcriptional factor, would bind to the EpCAM promoter. Yet they didn’t provide any experimental evidence. The author should at least use luciferase assay to test whether Slug binds to the EpCAM promoter and suppresses the expression.
Response: Thank you for your suggestion. In our previous study, luciferase assay and chromatin immunoprecipitation assay had demonstrated that Slug could bind to the proximal promoter region of EpCAM and inhibit the expression of EpCAM in cervical cancer cells [1]. Thereby, this part of result was cited in this article. Please do not hesitate to contact us and we will do our best to complete them as possible as we can. Thank you very much.
[1] Liu X, Feng Q, Zhang Y, Zheng P, Cui N. Absence of EpCAM in cervical cancer cells is involved in slug induced epithelial-mesenchymal transition. Cancer Cell Int. 2021;21(1):163.
- In the proposed model, Slug suppressed the translocation of b-catenin. However, based on the WB results, the amount of b-catenin decreased significantly in both WCL and nuclear portions. Slug likely suppressed the expression of b-catenin rather than its translocation into the nuclear.
Response: Thank you for your suggestion. We quite agree with your opinions. We also noted that the protein level of β-catenin decreased significantly in both WCL and nuclear portions under Slug overexpression. However, the expression of β-catenin at mRNA level did not have significantly change between Slug-overexpression cells and control cells. This was so interesting, and we suspect that for the reason of the reduction of translocation into the nuclear, the accumulated β-catenin in cytoplasm will degrade by the ubiquitination pathway. We already put our attention on this part, and we believe that we will finish this part of study in the near future. Please give our chance to show these results in our future study. Thank you very much. Please do not hesitate to contact us and we will do our best to complete them as possible as we can.
- Most of the experiments in this study are based on the overexpression of Slug in SiHa cells. I wonder about the expression level of Slug in cancer tissues compared to normal tissue. Would the cancer cell express less Slug? I also found a previous study reporting that Slug promotes breast cancer progression (PMID: 25652255). Does that mean that Slug's regulatory role differs in different cancer cells?
Response: Thank you for your suggestion. In our previous study[1], we found that: â‘ The Slug-positive rate decreased from normal human cervix (NC) samples (86.84%) to cervical cancer in situ (CIS) samples (62.50%) and invasive cervical cancer (SCC) samples (59.62%). â‘¡The immunoreactivity scores were also lower in CIS and SCC samples than in NC samples. â‘¢The average Slug protein level was lower in cervical carcinoma tissues than in normal cervix tissues.
Early studies showed that Slug exerts contrasting effects on cell proliferation and tumor growth among various cancers; for example, Slug promotes cell proliferation in lung cancer cells and glioblastoma cells [2,3], but inhibits cell proliferation in human epidermal keratinocytes and human prostate cancer cells [4,5]. Thereby, as your comment ‘Does that mean that Slug's regulatory role differs in different cancer cells’, we believe that in different cancer cells Slug present different regulatory effects. Thank you very much. Please do not hesitate to contact us and we will do our best to complete them as possible as we can.
- Cui N, Yang WT, Zheng PS. Slug inhibits the proliferation and tumor formation of human cervical cancer cells by up-regulating the p21/p27 proteins and down-regulating the activity of the Wnt/β-catenin signaling pathway via the trans-suppression Akt1/p-Akt1 expression. Oncotarget. 2016;7(18):26152-26167.
- Yang HW, Menon LG, Black PM, Carroll RS and Johnson MD. SNAI2/Slug promotes growth and invasion in human gliomas. BMC cancer. 2010; 10:301.
- Wang Y-P, Wang M-Z, Luo Y-R, Shen Y and Wei Z-X. Lentivirus-mediated shRNA Interference Targeting SLUG Inhibits Lung Cancer Growth and Metastasis. Asian Pacific Journal of Cancer Prevention. 2012; 13(10):4947-4951.
- Turner FE, Broad S, Khanim FL, Jeanes A, Talma S, Hughes S, Tselepis C and Hotchin NA. Slug regulates integrin expression and cell proliferation in human epidermal keratinocytes. The Journal of biological chemistry. 2006; 281(30):21321-21331.
- Liu J, Uygur B, Zhang Z, Shao L, Romero D, Vary C, Ding Q and Wu WS. Slug inhibits proliferation of human prostate cancer cells via downregulation of cyclin D1 expression. The Prostate. 2010; 70(16):1768-1777.
- I appreciate the model diagram in Figure 6. Yet some part of the model is hypothetical and without any experimental evidence. The authors should modify this diagram by adding references or labeling unverified parts in dash lines/ semi-transparent to avoid misinformation.
Response: Thank you for your suggestion. We are very sorry for our negligence in Figure 6. And the unverified parts had labeled in dash lines in Figure 6. Thank you very much. Please do not hesitate to contact us and we will do our best to complete them as possible as we can.

Reviewer 3 Report
In the submitted manuscript Liu et al. presented an extensive in vitro and in vivo study on Slug/EpCAM interactions in the regulation of stem-like phenotype of cervical cancer cells.
Although very robust, with plausible results and conclusion, the biggest drawback of this manuscript is a poor English language. There are lots of missing helping verbs and prepositions, wrong words were used, verbs were used instead of nouns and vice versa, as well as there are some too long and incomprehensive sentences. Therefore, an extensive scientific English language proofreading should be performed.
Other drawbacks:
1) All used cervical cancer cell lines should be mentioned in 'Abstract' because now it seems that you have used only SiHa cells.
2) Authors should inspect https://www.genenames.org/ and https://www.uniprot.org/uniprotkb and use only approved genes and protein symbols, taking account how to properly distinguish writing style between gene and protein symbols (https://www.biosciencewriters.com/Guidelines-for-Formatting-Gene-and-Protein-Names.aspx).
3) It is generally unclear what mean dilutions written as 101, 102, 103 and 104! In that light, rewrite Figure 2 legend because it is unclear what means "A. The numbers of 101", etc.
4) Application of statistical analyses is dubious:
- It is unclear why authors used parametric statistical tests, especially Pearson's corr. coeff., without testing first for normality of distribution.
- It is unclear what means "Mann–Whitney U-test and Student’s t-test (two-tailed) was used to analyses the univariate analysis"?!
- Correct phrase "presented as mean±SD" since some of bar graphs on Fig.3 and Fig.4 have only +SD error bar.
- Describe creation of Kaplan-Meier plots in 4.12. Statistical analysis section.
- State creator companies of used statistical software.
5) Line 344: You should not comment statistically unsignificant r (p = 0.0885)!
6) Provide used ImageJ version number and cite its proper reference (https://imagej.net/contribute/citing).
7) Provide qPCR cycling conditions. I suppose you actually used 2^-ΔΔCt method, so cite its reference PMID: 11846609
8) For the sake of reproducibility, provide actual models and manufactures of ALL used instruments, since constant referring to your previous papers is not convincing.
Author Response
Comments and Suggestions for Authors
In the submitted manuscript Liu et al. presented an extensive in vitro and in vivo study on Slug/EpCAM interactions in the regulation of stem-like phenotype of cervical cancer cells.
Although very robust, with plausible results and conclusion, the biggest drawback of this manuscript is a poor English language. There are lots of missing helping verbs and prepositions, wrong words were used, verbs were used instead of nouns and vice versa, as well as there are some too long and incomprehensive sentences. Therefore, an extensive scientific English language proofreading should be performed.
Response: Thank you for your suggestion. The English had been improved by using English editing from MDPI, Thank you very much. Please do not hesitate to contact us and we will do our best to complete them as possible as we can.
Other drawbacks:
1) All used cervical cancer cell lines should be mentioned in 'Abstract' because now it seems that you have used only SiHa cells.
Response: Thank you for your suggestion. We are very sorry for our negligence about this. The cell line which used in this study have added in abstract at line 30. Thank you very much. Please do not hesitate to contact us and we will do our best to complete them as possible as we can.
2) Authors should inspect https://www.genenames.org/ and https://www.uniprot.org/uniprotkb and use only approved genes and protein symbols, taking account how to properly distinguish writing style between gene and protein symbols (https://www.biosciencewriters.com/Guidelines-for-Formatting-Gene-and-Protein-Names.aspx).
Response: Thank you for your suggestion. We are very sorry for our negligence about this, the gene and protein symbols used in our manuscript have carefully checked and corrected (for example: Slug corrected to SNAI2, EpCAM corrected to EPCAM, etc.). Thank you very much. Please do not hesitate to contact us and we will do our best to complete them as possible as we can.
3) It is generally unclear what mean dilutions written as 101, 102, 103 and 104! In that light, rewrite Figure 2 legend because it is unclear what means "A. The numbers of 101", etc.
Response: Thank you for your suggestion. We are very sorry for our negligence about this. “101, 102, 103 and 104” have change as “10, 100, 1000 and 10000 cells”. Thank you very much. Please do not hesitate to contact us and we will do our best to complete them as possible as we can.
4) Application of statistical analyses is dubious:
- It is unclear why authors used parametric statistical tests, especially Pearson's corr. coeff., without testing first for normality of distribution.
Response: Thank you for your suggestion. We are very sorry for our negligence to describe this in our manuscript. Actually, the data that used for Pearson's correlation analysis have tested for normality of distribution by using GraphPad Prism (version 9.0). We have added this description at line 1950. Thank you very much. Please do not hesitate to contact us and we will do our best to complete them as possible as we can.
- It is unclear what means "Mann–Whitney U-test and Student’s t-test (two-tailed) was used to analyses the univariate analysis"?!
Response: Thank you for your suggestion. We are very sorry for our negligence about this. We have improved description about this in Statistical analysis section.Thank you very much. Please do not hesitate to contact us and we will do our best to complete them as possible as we can.
- Correct phrase "presented as mean±SD" since some of bar graphs on Fig.3 and Fig.4 have only +SD error bar.
Response: Thank you for your suggestion. We are very sorry for our negligence about this. We have corrected these in Figure 3 and Figure 4. Thank you very much. Please do not hesitate to contact us and we will do our best to complete them as possible as we can.
- Describe creation of Kaplan-Meier plots in 4.12. Statistical analysis section.
Response: Thank you for your suggestion. We are very sorry for our negligence about this. The description of Kaplan-Meier had added in Statistical analysis section. Thank you very much. Please do not hesitate to contact us and we will do our best to complete them as possible as we can.
- State creator companies of used statistical software.
Response: Thank you for your suggestion. We are very sorry for our negligence about this. The state creator companies of used statistical software had added in Statistical analysis section. Thank you very much. Please do not hesitate to contact us and we will do our best to complete them as possible as we can.
5) Line 344: You should not comment statistically unsignificant r (p = 0.0885)!
Response: Thank you for your suggestion. We are very sorry for our negligence about this. We have removed this comment. Thank you very much. Please do not hesitate to contact us and we will do our best to complete them as possible as we can.
6) Provide used ImageJ version number and cite its proper reference (https://imagej.net/contribute/citing).
Response: Thank you for your suggestion. We are very sorry for our negligence about this. The version number and cite of ImageJ have add in this manuscript. Thank you very much. Please do not hesitate to contact us and we will do our best to complete them as possible as we can.
7) Provide qPCR cycling conditions. I suppose you actually used 2^-ΔΔCt method, so cite its reference PMID: 11846609
Response: Thank you for your suggestion. We are very sorry for our negligence about this. The reference PMID: 11846609 have add in the part of Real-time PCR analysis. Thank you very much. Please do not hesitate to contact us and we will do our best to complete them as possible as we can.
8) For the sake of reproducibility, provide actual models and manufactures of ALL used instruments, since constant referring to your previous papers is not convincing.
Response: Thank you for your suggestion. We are very sorry for our negligence about this. The actual models and manufactures of instruments that used in this manuscript have add. Thank you very much. Please do not hesitate to contact us and we will do our best to complete them as possible as we can.

Round 2
Reviewer 3 Report
Authors have properly responded to all my concerns. However, there are still few thing which must be further clarified or corrected:
1) Sentence in lines 554-557 is still unclear:
- state which posthoc test was used with ANOVA
- explain what actually means "the χ2 test or one-way ANOVA was performed for comparisons among groups."
2) In lines 463 and 471 I suppose you actually meant "colonies", not large bowels or punctuation marks. :-)
Author Response
Authors have properly responded to all my concerns. However, there are still few thing which must be further clarified or corrected:
1) Sentence in lines 554-557 is still unclear:
- state which posthoc test was used with ANOVA
- explain what actually means "the χ2 test or one-way ANOVA was performed for comparisons among groups."
Response: Thank you for your suggestion. We are very sorry for our negligence to make a mistake in our statistical analysis section, we have corrected this mistake. Thank you very much for your suggestion on our statistical analysis section, it help us a lot to improve our manuscript. Thank you very much. Please do not hesitate to contact us and we will do our best to complete them as possible as we can.
2) In lines 463 and 471 I suppose you actually meant "colonies", not large bowels or punctuation marks. :-)
Response: Thank you for your suggestion. We are very sorry for our negligence about these mistakes, we have corrected these mistakes. Thank you very much. Please do not hesitate to contact us and we will do our best to complete them as possible as we can.
